# Large Coherent States Formed from Disordered *k*-Regular Random Graphs

**DOI:** 10.3390/e25111519

**Published:** 2023-11-06

**Authors:** Gregory D. Scholes

**Affiliations:** Department of Chemistry, Princeton University, Princeton, NJ 08544, USA; gscholes@princeton.edu

**Keywords:** coherence, expander graph, quantum resource

## Abstract

The present work is motivated by the need for robust, large-scale coherent states that can play possible roles as quantum resources. A challenge is that large, complex systems tend to be fragile. However, emergent phenomena in classical systems tend to become more robust with scale. Do these classical systems inspire ways to think about robust quantum networks? This question is studied by characterizing the complex quantum states produced by mapping interactions between a set of qubits from structure in graphs. We focus on maps based on *k*-regular random graphs where many edges were randomly deleted. We ask how many edge deletions can be tolerated. Surprisingly, it was found that the emergent coherent state characteristic of these graphs was robust to a substantial number of edge deletions. The analysis considers the possible role of the expander property of *k*-regular random graphs.

## 1. Introduction

How precisely engineered does a quantum system need to be in order to serve as a robust resource of large coherent states? Here, we study this question by characterizing the complex quantum states produced by mapping interactions between a set of qubits from structure in graphs. We study, in particular, maps based on *k*-regular random graphs where edges are randomly deleted. We calculate quantum states based on the eigenvectors of ensembles of these random graphs and explore the coherent delocalization of the states.

For an overview of coherence in the context of delocalization and dynamics of excitons and spins, the reader is referred to ref. [1]. Coherence can be classical, where, for example, it enables laser technology, or quantum mechanical, where, for example, it allows delocalization of wave functions. It comes from well-defined phase and amplitude relations where correlations are preserved over separations in space or time. In particular, coherence underpins the concept of interference. In terms of the density matrix, coherence is basis-dependent and the presence of coherence is indicated by off-diagonal entries in the density matrix for a state and can be quantified using various measures [2,3,4].

The kind of coherence that is discussed in the present work is coherent delocalization [5,6,7,8]. That is, we confine our focus to the single-excitation subspace of the relevant full tensor product space, consisting of states of *n* two-level systems. The relevant basis, of dimension *n*, is the set of product functions comprising (n−1) sites in the |0〉 state and one site, say *j*, in the |1〉 state, ${|0\rangle }_{1}{\ldots |1\rangle }_{j}\ldots {|0\rangle }_{n}$. The superposition states in this basis are delocalized states, where the extent of delocalization is often assessed by the inverse participation ratio, IPR,
(1)$$\mathrm{IPR}\left(\rho \right)=\sum_{i=1}^{n}\rho_{ii}^{2}.$$

Here, we use the measure purity, $\mathrm{Tr}\rho^{2}$, to compare the coherence of states systematically. Purity measures the mixedness of a state compared to purity = 1 for a pure state. We associate mixedness with less coherence. Purity is only useful as a comparative measure of coherent delocalization for systems of the same dimension. For more general comparisons, measures based on von Neumann entropy are preferred, as discussed below. For the comparisons made in the present paper, the measures are equally applicable. Various measures of coherence with regard to delocalization and entanglement are discussed by Smyth et al. [9]. Another useful paper is that of Levi and Mintert [4], which describes a quantitative assessment of coherent delocalization.

Let us say we have *n* qubits and can couple them pairwise. We represent the coupling scheme using a graph with *n* vertices and *m* edges, G(n,m), where each vertex designates a qubit (or atom, molecule, etc.) and pairwise interactions are drawn as edges connecting the coupled qubits. Here, we specialize this model so that every qubit is a two-level system with identical frequency gap, and each pairwise coupling is identical—set to a value of unity. The eigenvectors and eigenvalues of the adjacency matrix of the graph G(n,m) characterize the corresponding set of pure states.

The adjacency matrix A(n,m) is the n×n Hermitian matrix containing (in the case of an undirected graph) values of 1 at each i,j entry, where vertex *i* is connected by an edge to vertex *j*. By constructing the density matrix ρ for an ensemble of eigenstates of A(n,m), we obtain a general mixed state representing the emergent state (see below for the definition) of the ensemble of graphs G(n,m). We are looking for properties of this quantum state ρ and how they relate to the structure imposed by the graphs G(n,m). The present work extends a prior study [10] in a new direction.

The present work is motivated by the possible roles that coherent states and purity might play as quantum resources [11,12,13,14,15,16,17]. According to quantum resource theories [18], the ingredients needed for quantum operations are free states, free operations, and resources. Free states are generally states that are readily available. Similarly, free operations, which map free states to free states, are easily implementable. The crucial ingredients are resources. One systematic framework for quantifying coherence in the context of a resource theory is reported by Baumgratz, Cramer, and Plenio [3]. Here, the merits of distance-based metrics are discussed. One such measure is quantum relative entropy $S_{\rho ∥\sigma }$, which quantifies the distance of a state ρ from the nearest incoherent (mixed) state σ:(2)$$S_{\rho ||\sigma }=\mathrm{Tr}\left(\rho \log_{2}\rho \right)-\mathrm{Tr}\left(\rho \log_{2}\sigma \right).$$The state σ is constructed by setting the off-diagonal entries from the density matrix of ρ to zero. The relative entropy, therefore, explicitly measures how far a state is from a comparable mixed state, meaning, for our purposes, how coherent the state is compared to its incoherent analog.

In practice, coherent states likely have attributes that could be used for functions in ways that make use of interference phenomena or that allow other non-classical correlations to be exploited. Such applications include sensors, detectors, switches, communications, or transport involving excitons and charges [19,20,21,22,23]. A concrete example of how we might design simple organized networks that utilize interference phenomena is reported by Fassioli et al. [24].

A challenge for producing coherent states for resources is that large, complex systems are often fragile—that is, coherent states tend to be quite localized [1,25,26]. On the other hand, the emergent phenomena in classical systems become more robust with scale. A requirement for any large and complex quantum system, whose structure is, in a sense, ‘organic’, is that it is robust to various kinds of disorder. This is where the concept of emergent or collective phenomena [27,28] might be important.

An emergent phenomenon is manifest when numerous small interactions add coherently to produce a stunning collective effect that appears above some threshold of the number of coherently interacting systems. There are many examples in classical systems that are produced by synchronization [29,30,31,32,33,34,35,36,37]. One illustrative example is the report by Buhl et al. [38] where the motion patterns of desert locusts were studied. It was observed that at la ow density of the insects, they move independently—as individuals. However, at a critical density, their motion becomes synchronized, and huge swarms march collectively through crop fields, leaving devastated vegetation in their wake. In another intriguing example, Strogatz et al. [39] discuss how the natural vibrations of a huge concrete and steel footbridge, the Millennium Bridge across the Thames river in London, are amplified by the synchronized footsteps of pedestrians. As remarkable as it sounds, it was found soon after the bridge opened that once the density of pedestrians on the bridge exceeds a threshold, people tend to walk in lock-step. The many small exertions from the walkers add coherently and amplify the natural vibrations of the bridge so that it sways side-to-side.

In quantum systems, an emergent state is signaled by a single eigenvalue that splits away from the numerous other eigenvalues in the spectrum, opening up an energy gap [28]. An example is shown later in this paper, in Figure 3. Can we make progress by considering networks structured so that they host emergent states? What are some properties of such emergent states of large quantum systems?

There is also interest in assessing the possibility that non-classical aspects of coherence could be used for biological functions. Clearly this would be manifest in a very different way than, for example, protocols for quantum circuits, and the key open question is how quantum effects could feasibly be implemented in such complex and disordered systems—what would be the conceptual and organizational frameworks? It is likely that, if feasible, the function or sensing based on complex coherent states would need to be fairly simple and certainly resilient to variations in the quantum system, which motivates the present study, where some very basic ideas for the foundation of this program are developed.

## 2. Methods

The structure of G(n,m), which determines how qubits interact with each other, defines $2^{n}$ quantum states in the Hilbert space $H=H_{1}⊗H_{2}⊗\cdots ⊗H_{n}$. Using the graph model, we can systematically build the full set of *N*-excitation subspaces of H (unpublished). Here, we specialize to the single excitation subspace—where only a single qubit in any product state is in the |1〉 configuration. These states represent a set of coherent states [11,40,41], analogous to the molecular (Frenkel) exciton states [42,43]. The general states ρ we study are constructed from the ensemble of eigenstates associated with the largest eigenvalue of the graph adjacency matrix (the emergent state). We can thereby look for correlations between the structure on H, encoded by G(n,m), and properties of the state ρ.

We produce the G(n,m) from *k*-regular random graphs by random edge deletions until *m* edges remain. A *k*-regular random graph on *n* vertices has a total of kn/2 edges, arranged such that each vertex connects to precisely *k* edges. One such graph is randomly generated, then edges are deleted at random to produce G(n,m). For each (n,m,k), 1000 graphs are generated to construct ρ. Some examples of small graphs are shown in Figure 1 to give a sense for how they evolve as a function of *m*.

It is well known that when *n* is large and m≪n/2, a random graph comprises small disconnected trees, but above a threshold, m≫n/2, the graph is completely connected [44,45,46]. Here, read ≪ and ≫ to be ‘sufficiently’ less than and greater than, respectively. You can see this phase transition even for the small graph drawn in Figure 1. More specifically, a general random graph undergoes various phases as the number of edges m(n) increases from 1 to n2, the number of edges *m* in the complete graph on *n* vertices. The ‘early’ phases, for small *m* compared to *n*, are very similar for *k*-regular graphs, provided that *k* is not too small. The first phase is delineated by m(n)=o(n) and manifest as graphs that comprise exclusively trees, almost surely as n→+∞. That is, G(n,m) is a forest.

We have to be careful interpreting the graphs with m=30, shown in Figure 1, because *n* is small so as to make the picture clearer. Note also that the graphs are drawn with the largest connected components in the center, which tends to emphasize those components somewhat. With those caveats, these pictures show graphs in the second phase, where m=cn/2 for some positive constant *c*. As *c* increases from c<1 to c>1, the small trees from the first graph phase grow smoothly until a point where adding edges simply connects the subgraphs [47], quite rapidly producing a single connected graph, such as those shown in Figure 1 for m=60. Connected subgraphs are more likely to recruit edges, simply because they contain more vertices. Even the simplest 3-vertex tree is three times more likely to obtain a new edge than an isolated vertex. This phase transition to produce the single connected graph turns out not to be particularly significant for the focus of the present study because the threshold for a weakly mixed quantum state occurs only after significantly more edges are added to the graph.

Examinations of synchronization on classical networks, including those represented by random graphs [48], are relevant to the present study, as noticed in the prior work [10,49,50]. There, we studied networks of qubits with a fixed coupling structure (i.e., G(n,m)) but with a distribution of frequencies for qubits over the vertices. Those calculations could be compared directly with simulations of classical phase oscillator synchronization [30,31,48,51,52], and it was observed that the order parameter of the classical system relates to the inverse participation ratio of the emergent quantum state. The ensemble of networks studied in the present work is different because there is no frequency disorder. Instead, there is a distribution of structure in the network, manifested by edges missing from the base *k*-regular graphs. This structural disorder introduces an off-diagonal disorder in the ensemble of adjacency matrices—which, thus, have the properties of a special kind of Bernoulli random matrix where off-diagonal entries are either 0 or 1.

## 3. Results and Discussion

Here, we examine quantum states derived from the graphs G(n,m) described above. The states are constructed from an ensemble of eigenvectors corresponding to the largest eigenvector of the adjacency matrix of each graph in the ensemble. This is the emergent coherent state, where each eigenvector is $\left(1,1,1,\ldots \right)/\sqrt{n}$ in the limit of a pure state. The key feature of our quantum system is that we know that we have *m* edges, but we do not know which vertices are coupled. As long as *n* is sufficiently large, the results are not sensitive to the size of our system (i.e., to *n*). The representative results are shown in Figure 2. By ‘representative’, we mean that similar results for systems with various numbers of vertices were calculated and they are all qualitatively consistent with the data selected for the figure.

The purity of ρ, defined as $\mathrm{Tr}\rho^{2}$, and the relative von Neumann entropy, defined as the entropy distance from ρ to the nearest fully mixed state [11], see Equation (Equation 2), follow comparable trends. The abscissa of these plots is denoted ‘fraction of edges’, meaning that *m* is indicated as a fraction of the maximum possible edges, which is equal to nk/2. The notable result of our calculations is that the purity (or, similarly, relative entropy) of ρ rises steeply as the graph becomes more dense. Then, when more than half the allowed edges are in place, the purity (relative entropy) plateaus. Thus, even with half the edges missing, the state represented by k=40 graphs has a purity of >0.95.

What is the mixed state corresponding to the largest eigenvalue of the graph ensemble, and what determines the transition to an almost pure state when sufficient edges are included? The distributions of the number of edges at each node are approximately normal, with an average degree *d* consistent with the expectation of d=2m/n. For k=20, 40, 80, and 160, we find at the threshold for purity ≥0.95 degrees of 14, 20, 26, and 32, respectively. It is evident that it is not simply the average degree *d* that is important but also how the edges are distributed across the network of vertices. A more uniform distribution for any fixed *m* is better guaranteed when *k* is smaller, because edges cannot excessively accumulate on a small set of nodes.

A uniform distribution of *m* edges across *n* vertices means that the average graph, that is, where every vertex has *d* edges, is *d*-regular. The eigenstate with the greatest eigenvalue, then, is (1, 1, 1, …) (not normalized). The graphs in the ensemble that are disordered exhibit a distribution of coefficients, which reduces the purity, as seen in Figure 3. When the number of edges is small enough, some of the coefficients are zero. Otherwise, they are distributed such that as m→kn/2, the mean of the distribution tends to 1 and the variance goes to zero. For example, for the graphs shown in Figure 3, the mean coefficient is calculated to be 0.865, 0.947, and 0.969 for m=1000, 2000, and 3000, respectively.

While the eigenvector coefficients are distributed around the expected values for a *k*-regular random graph (with k=d), the eigenvalue, $\lambda_{0}$, does not show significant distribution in the spectrum. This is because for any classical random graph G(n,p), where *p* is the probability for an edge, the largest eigenvalue is almost surely [1+o(1)]np when n≫logn. For graphs generated as G(n,m), np≈2m/n. That is, the eigenvalue $\lambda_{0}$ should be equal to the average degree, regardless of the graph. Chung and co-workers [53] considered a special random graph model where edges are assigned independently to each pair of vertices, which allows edges to be distributed according to weights or rules. For these graphs they find that the largest eigenvalue of the adjacency matrix is bounded by the variance of the vertex degree distribution. We see a consistent trend in the graphs analyzed in the present work, evident in the distributions plotted in the insets of Figure 3.

The remaining eigenvalues are clustered within a distribution [54] that approaches Wigner’s semicircle law for the spectrum of a random matrix, with the limit distribution supported on the interval $\left[-2\sqrt{d-1},2\sqrt{d-1}\right]$.

The spectra as a function of *m*, or specifically, the second eigenvalue, are also compared with eigenvalue bounds known for expander graphs—and in particular, *k*-regular random graphs. Expander graphs [55,56,57] are sparse graphs—that is, they are far from complete graphs—yet, they are highly connected. That is, for any vertex subset *X* of size *x*, of less than half the vertices in the graph, there are at least ϵx edges to vertices in the set *X* coming from vertices outside of *X*. Expanders have many desirable properties, such as being an efficient network for random walks. *k*-regular random graphs are examples of expanders, and for these graphs Alon and Boppana [58] conjectured a bound on the second largest eigenvalue of the adjacency matrix, such that [59] $\lambda_{1}\leq 2\sqrt{k-1}-o_{n}\left(1\right)$.

In Figure 4 we plot the calculated gap $\lambda_{0}-\lambda_{1}$ compared to that predicted by the Alon–Boppana bound for a *d*-regular random graph, where *d* is the expected degree d=2m/n. We compare the data for n=400, k=20 and n=400, k=80. The predicted eigenvalue gap matches the calculated gap remarkably well, suggesting that, on average, the ensemble state based on graphs with *m* edges is reasonably represented by an ensemble of (2m/n)-regular random graphs, in particular, at larger values of *m*, where we found that the eigenvector coefficients are close to the expected (1,1,1,…) values. The mild dependence of the results shown in Figure 2 on *k* is, therefore, likely due to low values of *k* ‘forcing’ a *d*-regular average graph, as speculated above.

## 4. Conclusions

It was shown that large networks of qubits can provide an emergent state that serves as a robust coherent state. A modest number of random interactions among the qubits is sufficient for a relatively pure state, provided that the distribution of interactions is directed using a rule. In this case, the rule was that the perfect graph is a *k*-regular random graph. By perfect graph, here, we mean the graph with no missing edges. Even with half the edges missing from the perfect graph, the states showed high purity, which was derived from the way the graphs take the properties of the expected degree. In other words, removing random edges from a *k*-regular random graph produces a random graph that has properties very similar to a *d*-regular random graph, where d<k is the expected degree of the new graph. The coherent states are, thus, very resilient to disorder in the network; the connections among the nodes do not need to be precisely positioned but can be added at random. By the resiliency of coherent states to disorder, the idea is that there is a large variation in the structure of the states within the ensemble that are averaged to give a state of interest (the density matrix). Specifically, this ensemble includes states produced by numerous different ways that the vertices can be connected; however, the purity of the state is high, and it, therefore, does not reflect that disorder in the graph edge patterns. Moreover, we can delete or add many edges at random, and still the coherence of the state, indicated by its purity, remains almost constant. We should note that while the states discussed in the present work have high purity and, therefore, coherence, they are not well suited for quantum engineering or computing applications because the states are not well defined. They cannot be systematically processed in an appropriate way. The value of such states might be in using their coherence in different ways, as speculated in the introduction.

## Figures and Tables

**Figure 1 entropy-25-01519-f001:**
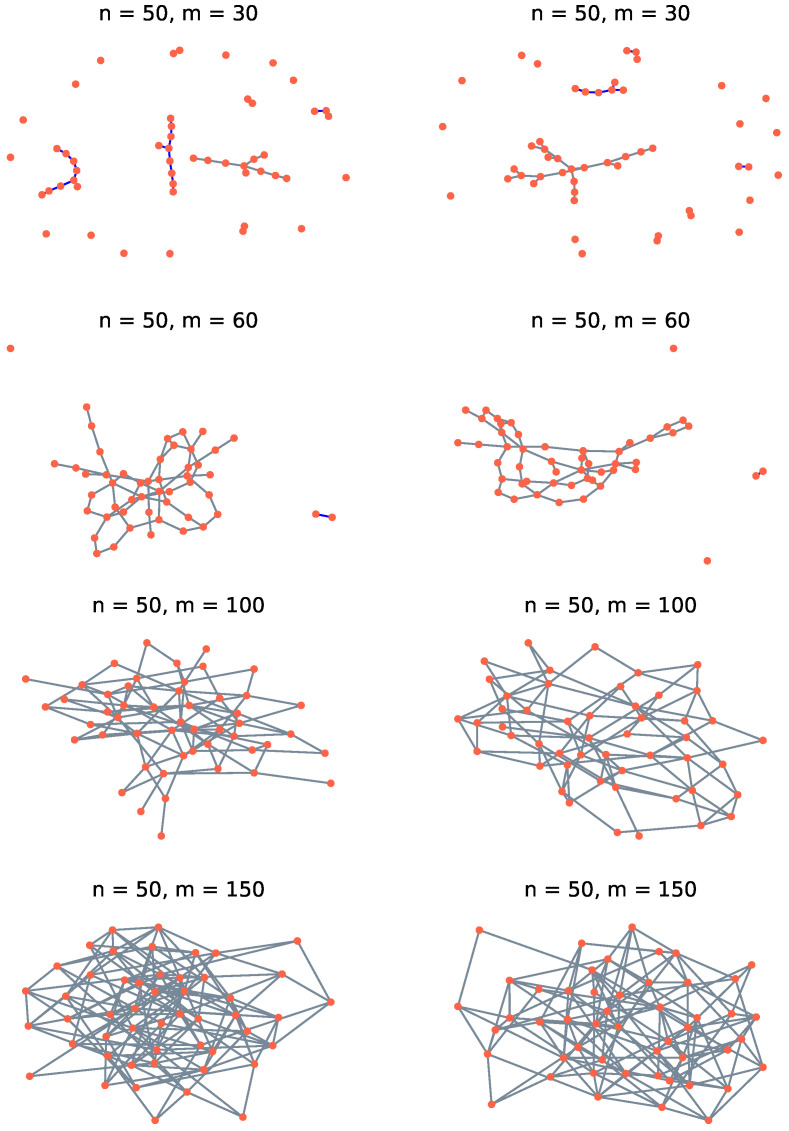
Drawings of small *k*-regular random graphs for 50 vertices (n=50) and various numbers of edges (*m*). Here k=10.

**Figure 2 entropy-25-01519-f002:**
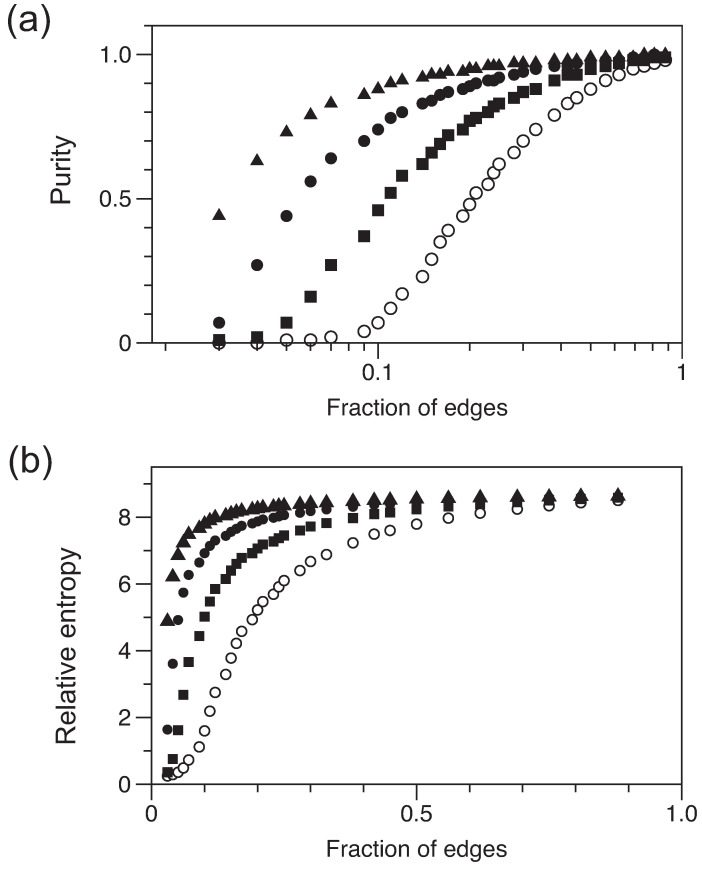
Characterization of the states ρ as a function of the number of edges added to the underlying graphs. There are 400 vertices, and values of *k* increase from the lower curve (open circles) to the upper curve (triangles) as k= 20, 40, 80, and 160. (**a**) Purity, with a logarithmic scale for the *x*-axis. (**b**) Relative von Neumann entropy, with a linear scale for the *x*-axis.

**Figure 3 entropy-25-01519-f003:**
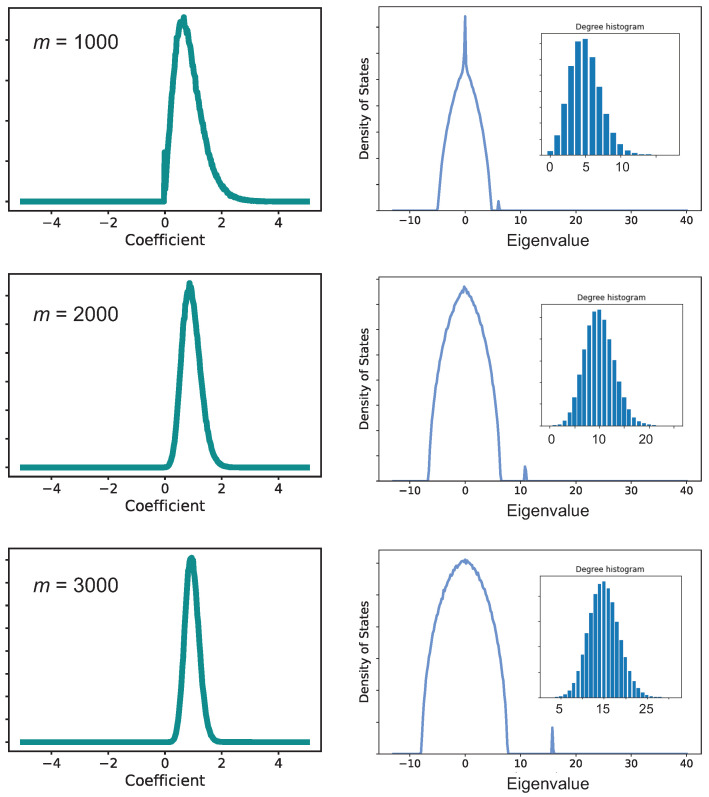
Data from calculations for n=400 and k=80 with various values of *m*, as indicated. The left panels plot the distribution of eigenvector coefficients, normalized such that the emergent state has an eigenvector (1,1,1,…). Notice that the emergent state becomes dominant for large values of *m*. The right panels show the spectrum of the eigenvalues. The emergent state splits off from the semi-circle distribution to higher eigenvalues. The inset panels show the distribution of degree at the graph vertices.

**Figure 4 entropy-25-01519-f004:**
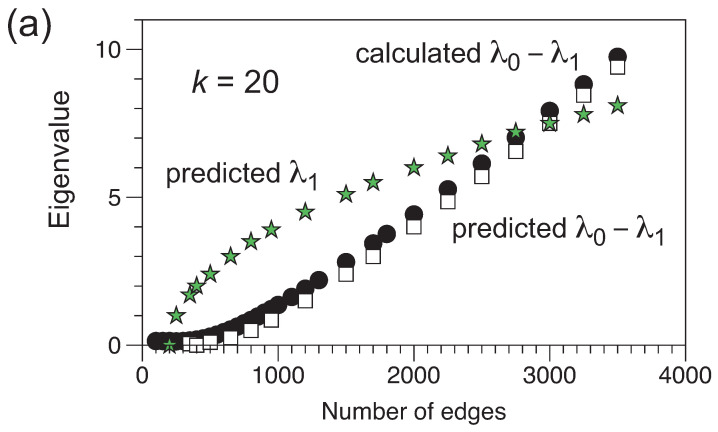
The difference in value between the first and second eigenvalues, $\lambda_{0}-\lambda_{1}$ for n=400 as a function of *m*. Results of the numerical calculations are indicated by the black circles. The Alon–Boppana bound for $\lambda_{1}$ is indicated by the stars. Taking $\lambda_{0}=2m/n$, the average degree, $\lambda_{0}-\lambda_{1}$ according to the Alon–Boppana bound is shown as the squares. (**a**) Results for k=20. (**b**) Results for k=80.

## Data Availability

The data are all calculated and can be reproduced using the methods described in the paper.

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
