# Peer review of "Large Coherent States Formed from Disordered k-Regular Random Graphs"

_entropy, 2023, doi:10.3390/e25111519_

Round 1

Reviewer 1 Report

Comments and Suggestions for Authors

The manuscript “Large Coherent States Formed from Disordered k-regular Random Graphs” investigates the resilience of large quantum systems to significant disturbances, here the random deletion of interaction channels. In this work, the interactions are determined by k-regular random graphs, where the interaction channels are described by their edges. Particularly, I find it quite unexpected that the deletion of as many as 50% of the edges leads to rather minute changes in purity and relative entropies for a large k.

The manuscript is original and well-motivated. The results seem interesting, and they definitely warrant publication in some form. The Entropy Journal seems to be an appropriate venue for this work. 

I have a few requests for the author, however, mainly to improve the readability of the manuscript for a broader audience. 

1) It appears that the author refers to states with even minute coherence as coherent. Under this definition, a class of coherent states is quite broad; however in other sub-fields of quantum science “coherent states” (e.g. quantum optics) can be very restrictive. In addition, I was not able to find a clear definition of a “coherent state” in prior literature, e.g. where quantum coherence is quantified as a quantum resource. It would help to introduce quantum coherence and give the definition of a coherent state in 1-2 sentences in the introduction.

2) While purity and relative entropy are defined, a causal reader might lack operational understanding of the practical significance of those measures. For example, it is commonly understood that fidelity is a good measure of nonclassicality. On the other hand, even small deviations from unity fidelity require significant error correction efforts in quantum information applications. In light of the above, it would help the reader to best appreciate the result if some discussion on advantages and disadvantages of purity and relative entropy as characterization measures.

3) In Fig. 2, “representative” results are shown. Would it make sense to depict all the results by, for instance, depicting the standard deviation of purity and entropy values as error bars?

4) I do not fully understand the statement on resiliency of coherent states to disorder made in the conclusion. While it is evident from this work that states exhibit high purity after edges are removed at random, can anything be said (or speculated) on the need for quantum error correction should those states have been used for quantum information processing?

5) (minor point) 4-th line of abstract: please correct the typo, I assume “thin” should read “think”?

In conclusion, the paper is well-written, and I support its publication. I think the readers will benefit if the points above are addressed prior to publication.

Author Response

Please find author's replies in the attachment.

Reviewer 2 Report

Comments and Suggestions for Authors

In the present paper, the author addresses the question of quantum state robustness when a large complex system has some connections randomly broken. He examines the situation of the k-regular random graphs with some edges randomly deleted. He shows that if even many edges are deleted (13000 out of 16000 for the lowest panel of Figure 3), the emergent quantum state still dominates. I believe that this is a very important analysis, and the paper merits publication.

I have a couple of general recommendations for making the paper more attractive to a broad audience. I would suggest extending the introduction to provide the readers with more information about quantum resources, emergent phenomena, etc. I would also suggest adding more discussions of Figure 1. What is the value of k for this example? How frequently is the situation shown in the topmost panel, when many vertices are completely disconnected, but some have two or even three connections?

One more small thing: the last three sentences of the abstract are repeated in the first paragraph of the Introduction. I would suggest rewriting this.

Author Response

(The authors gave the same response as above.)
